# Clock Genes Profiles as Diagnostic Tool in (Childhood) ADHD—A Pilot Study

**DOI:** 10.3390/brainsci12091198

**Published:** 2022-09-05

**Authors:** Alexander Dück, Olaf Reis, Henrike Wagner, Katja Wunsch, Frank Häßler, Michael Kölch, Mariana Astiz, Johannes Thome, Christoph Berger, Henrik Oster

**Affiliations:** 1Department for Child and Adolescent Psychiatry and Neurology, Rostock University Medical Center, 18147 Rostock, Germany; 2GGP Group, 18059 Rostock, Germany; 3Institute of Neurobiology, University of Luebeck, 23562 Luebeck, Germany; 4Circadian Physiology of Neurons and Glia Laboratory, Achucarro Basque Center for Neuroscience, 48940 Leioa, Basque Country, Spain; 5Department for Psychiatry and Psychotherapy, Rostock University Medical Center, 18147 Rostock, Germany

**Keywords:** ADHD, clock genes, melatonin, cortisol, actigraphy

## Abstract

Attention deficit hyperactivity disorder (ADHD) is a very common disorder in children and adults. A connection with sleep disorders, and above all, disorders of the circadian rhythm are the subject of research and debate. The circadian system can be represented on different levels. There have been a variety of studies examining 24-h rhythms at the behavioral and endocrine level. At the molecular level, these rhythms are based on a series of feedback loops of core clock genes and proteins. In this paper, we compared the circadian rhythms at the behavioral, endocrine, and molecular levels between children with ADHD and age- and BMI-matched controls, complementing the previous data in adults. In a minimally invasive setting, sleep was assessed via a questionnaire, actigraphy was used to determine the motor activity and light exposure, saliva samples were taken to assess the 24-h profiles of cortisol and melatonin, and buccal mucosa swaps were taken to assess the expression of the clock genes *BMAL1* and *PER2*. We found significant group differences in sleep onset and sleep duration, cortisol secretion profiles, and in the expression of both clock genes. Our data suggest that the analysis of circadian molecular rhythms may provide a new approach for diagnosing ADHD in children and adults.

## 1. Introduction

Attention deficit hyperactivity disorder (ADHD), with a prevalence of 5.29% (95% CI = 5.01–5.56) in childhood and adolescence [1] and 3.4% (no CI provided) in adulthood [2], is one of the most common neurodevelopmental disorders globally. Sleep disturbances are widely studied in ADHD. Studies using polysomnography (PSG) and actigraphy have shown significant connections of ADHD with various sleep disorders in childhood and in adulthood [3,4,5,6]. The role of sleep and circadian rhythmicity in the pathophysiology of ADHD itself is a matter of debate and controversy [7,8]. In the search for markers (e.g., for the therapeutic success of stimulant medication), objective measuring instruments such as actigraphy have proven to be useful and effective [9]. The core symptoms of ADHD—inattention, impulsivity, and motor hyperactivity—are very similar to those seen after sleep deprivation. Several studies have described a shorter average sleep duration in children with ADHD [10], chronic sleep-onset insomnia, and a diurnal preference toward eveningness was found in children and adults with ADHD [10,11,12,13]. However, the results remain inconsistent, even in studies using objective tools such as PSG and acitgraphy. The emotional comorbidities or medication in the subjects and controls are discussed as important confounding factors [7].

The circadian clock system is responsible for adapting physiological processes and behavior to the 24 h day–night rhythm. It is a major driver of sleep–wake rhythms [14,15]. The main Zeitgeber of the clock is light. The suprachiasmatic nucleus of the hypothalamus (SCN) acts as the master circadian pacemaker. From SCN, light-entrained oscillations are autonomously transmitted to different cells in the brain and in the periphery [16].

The SCN is responsible for regulating the rhythmic secretion of various hormones such as cortisol and melatonin. Alterations of these secretions have been described in ADHD in children and adults, but with partially inconsistent results [13,17,18,19].

At the molecular level, circadian rhythm regulation works through interlocked positive and negative transcriptional–translational feedback loops of a set of core clock genes and their protein products. For example, heterodimers of CLOCK and BMAL1 activate the transcription of the negatively-regulating *PERIOD*-genes (*PER1*, *PER2*, and *PER3*). The PER proteins then inhibit the transcriptional activation by CLOCK and BMAL1. This results in a circadian rhythmic oscillation of *PER* transcription [20,21,22]. Single-nucleotide polymorphisms (SNIPs) of clock genes have been studied in the context of ADHD in children and adults, with inconsistent results [23,24,25].

Baird et al. (2012) examined circadian rhythmicity at the molecular, endocrine, and behavioral levels in adult ADHD for the first time, and found disturbed rhythmicity at all levels [20]. Thus far, the mechanisms described here were only examined for adults. We hypothesized, by analogy, that they would also apply for children. Given that this hypothesis would be validated, we further hypothesized that differences in circadian rhythmicity on behavioral, endocrine, and molecular levels found between the groups could be used as diagnostic markers. In other words, we assumed that plausible differences in circadian rhythmicity on the behavioral, endocrine, and molecular levels produce indications for the design of a bigger pilot-study to follow, wherein constructs should be tested for small, but more sufficient samples, suitable for the estimation of the effects sizes. Thus, the hypotheses presented here are not of an affirmative nature, but rather guide our exploration of the data.

## 2. Materials and Methods

### 2.1. Subjects and Ethics

Ethical approval for the conduct of the study including the use of all human tissues was given by the ethics committee of Rostock University (Registration-number: A2012-0116), and written informed consent was obtained from each study participant and caregiver. The study was conducted according to the ethical guidelines of the Declaration of Helsinki.

We used a head to head comparison design with two groups: a group of ADHD patients and a non-affected group (controls). The ADHD sample was recruited at the Department for Child and Adolescent Psychiatry and Neurology, Rostock University Medical Center. Inclusion criteria were an ADHD diagnosis according to ICD 10 [26] F 90.x, male gender, age from 8 to 12 years, absence of physical problems, no psychiatric comorbidity, no current or past medication, no epileptic activity. The healthy controls were recruited at a local football camp. Exclusion criteria for both groups were any other psychiatric diagnosis than ADHD, any mental or physical problem, IQ below 85, any epileptic activity, and any current or past medication. Additionally, the diagnosis of ADHD was not allowed in the control group. Diagnostic assessment included an extensive exploration of proband and family by an experienced child and adolescent psychiatrist, K-Sads-Interview [27], an electroencephalogram (EEG), and a physical examination.

### 2.2. Questionnaire

The Children’s Sleep Habits Questionnaire (CSHQ) is a parent-report screening survey published by Owens et al. in 2000, which has been validated for school-aged children [28]. It has been translated into many languages [29,30,31] and used in multiple studies on children of different ages and neuro-developmental conditions [10,32,33]. In our study, we used the German version translated and validated by Schlarb et al. (2010). The internal consistency of the German version was α = 0.68 for the entire questionnaire and ranged from α = 0.23–0.70 for the subscales. The retest-reliability of the German version was reported to be r = 0.76 [31]. The 35 items of the questionnaire are grouped into subscales: Bedtime Resistance, Sleep Onset Delay, Sleep Duration, Sleep Anxiety, Night Awakenings, Parasomnias, Sleep-Disordered Breathing, and Daytime Sleepiness. The Total Sleep Disturbance score includes all items with a cut-off total score of 41 and a sensitivity of 0.80 and specificity of 0.72 [28]. The primary tool for establishing the diagnosis of ADHD was the clinical assessment as it is manualized within the K-Sads [27]. Moreover, we collected dimensional ratings from parents with the FBB-ADHS [34], a questionnaire that allows to differentiate between subtypes of ADHD. However, as this differentiation would have divided the small sample into even smaller groups and because the main theme of this study was to compare the ADHD and control groups with each other, we refrained from analyzing the subtypes of ADHD.

### 2.3. Actigraphy

The actigraph used was a wrist-worn sensor resembling a small watch (MotionWatch 8, CamNtech Ltd., Manor Farm, Fenstanton, Cambridgeshire, UK). Participants wore the actigraph all day for at least 5 consecutive days on the wrist of the non-dominant hand. For the study protocol, see Figure 1. Motor activity was measured by a tri-axial Microelectromechanical systems (MEMS) based accelerometer with a range of 0–8 g and a sensitivity of 0.1 g. Light exposure was measured with a light sensor with a range up to 64,000 lux and with 0.25 lux resolution included in the device. The actigraphs were waterproof, so they did not have to be removed while showering or bathing. The actigraphs were calibrated by an experienced medical technical assistant. We used the same settings in both groups. The collected data were stored on a laptop computer and later transferred to a database under consideration of data security. The light exposure and movements were summarized and averaged in 1 min bins. Inter-daily stability and intra-daily variability were computed directly from the raw activity data according to the method used by Witting et al. [35]. The intra-daily variability stands for the hour-to-hour changes in activity and describes the transitions between rest and activity. It indicates the daytime resting or nighttime activity. Inter-daily stability scaled from 0 to 1 stands for the consistency of the circadian patterns (light exposure, motor activity) from one day to the next. This indicates the strength of correlation between a 24 h rhythm with an environmental Zeitgeber. Relative amplitude was identified as a variable that does not rely on the assumption of a cosine fit. This is based on the difference between the most active 10 h interval (M10) and the least active 5 h interval (L5) of the day [36].

### 2.4. Melatonin and Cortisol Analysis

The hormones melatonin and cortisol were determined from the saliva samples. These were collected in parallel oral mucosa samples every 4 h, with an additional collection at 10 rpm to map the dim-light melatonin onset (DLMO; see Figure 1). The subjects spat through a straw into a small container (Sarstedt Salivette). Nighttime samples were collected under the dim light condition. The samples were then stored −20 °C until they were transferred into the Laboratory for Molecular Psychiatry at Rostock University Medical Center. Both salivary cortisol and melatonin were assayed by ELISA (IBL International, Hamburg, Germany).

### 2.5. Clock Gene Expression Analysis

The 24 h transcription profiles of the clock genes *BMAL 1* and *PER 2* were determined by quantitative PCR on cDNA from the total RNA preparations. For this purpose, cells of the buccal mucosa were removed using a sponge. Samples were taken every 4 h, as shown in Figure 1. These samples were immediately frozen at −70 °C, stored and sent to the laboratory of the Institute for Neurobiology at Luebeck University for analysis. RNA was isolated with TRI Reagent (Ambion, Austin, TX, USA, Catalogue number 15596018). Briefly, brushes were carefully taken out form the tube and the buccal mucosa cell suspension was centrifuged at 14,000 rpm, 4 °C, 30 min. The supernatant was discarded and the samples were incubated at 65 °C for 15 min with shaking (400 rpm). One mL of the TRI reagent was added and the samples were incubated for 5 min at room temperature (RT). Chloroform (Sigma, St. Louis, MO, USA, Cat no. C2432) was added (200 μL) and after vigorous mixing, the samples were centrifuged at 14,000 rpm for 15 min at 4 °C. The upper phase was transferred into a fresh tube. Pre-chilled isopropanol (500 μL, Carl Roth Cat no. AE73.1) and 1 uL Glycoblue (Thermofisher, Waltham, MA, USA, Cat no. AM9515) were added to each sample followed by a 30 min incubation on ice. Samples were centrifuged at 14,000 rpm for 30 min at 4 °C and the supernatant was removed. The RNA pellet was washed with 1 mL pre-chilled ethanol (70%) by centrifuging the samples at 14,000 rpm for 10 min at 4 °C. The pellet was dissolved in 12 μL of RNAase-free water. cDNA synthesis was performed taking 10 μL of the RNA/sample and using the High Capacity cDNA Reverse Transcription Kit (Thermofisher, 4368814) following the manufacturer’s instructions. The retrotranscription was performed by incubating the sample at 25 °C for 10 min, at 37 °C for 120 min, and the last incubation step at 85 °C for 5 min. cDNA was diluted to 1:4 and stored at −20 °C until analysis. qPCR was performed using the Go-Taq qPCR Master Mix (Promega, Madison, WI, USA, Cat. No A6002) with a CFX96TM thermocycler (Bio-Rad, Herakles, CA, USA) using the following program (3 m at 95 °C, followed by 40 cycles of 15 s at 94 °C, 15 s at 60 °C, and 20 s at 72 °C). All samples were loaded in duplicate. Data analysis was performed using the ΔΔCT method [37] after confirmation of comparable amplification efficiency for every target gene (PER2 and BMAL1) and the house-keeping gene (human β2-Microglobulin (B2M). The primer sequences used were: B2M (fwd): GCCGTGTGAACCATGTGACT and (rev) GCTTACATGTCTCGATCCCACTT. BMAL1 (fwd) GTACCAACATGCAACGCAATG and (rev) TGTGTATGGATTGGTGGCACC. PER2 (fwd) GCCAAGTTTGTGGACTTCCTG and (rev) CTTGCACCTTGACCAGGTAGG.

### 2.6. Statistical Analysis

Given the small sample size, we refrained from the assumption of normal distribution and used the non-parametric Mann–Whitney-U-test to compare FBB-ADHS, CSHQ variables, IQ, age, weight and height as well as the IV, IS, M10, and L5 between groups. The significance level was set to *p* ≤ 0.05. In order to describe the circadian rhythm of the genetic expression and the hormone data, we performed a cosinor fit using the software Time Series Analysis Serie Cosinor 6.3 (http://www.euroestech.com accessed on 10 February 2014). We used the cosinor2 toolbox in R (Version 4.0.3 [38]) to compare the population mesors, amplitudes, and acrophases as introduced by Bingham et al. [39]. Notably, if the acrophases of two populations are significantly different, the results of the amplitude difference test are not reliable and should not be interpreted. In order to test whether the proportion of the variance explained by the rhythm (percent-rhythm) was significantly different between the groups, we used the Fisher-z test [40] and compared the coefficients of correlation between the observed and fitted values. The percent rhythm was defined as the squared correlation coefficient. Various chronometric parameters can be generated from the cosinor curve and used for further statistical analysis: the amplitude is the maximum extent of the oscillation. The acrophase is the timepoint of the peak of the rhythm and the mesor is defined as the middle value of the cosine wave. The pilot study presented here refrained from estimating the effect sizes in group differences as its primary goal was to inspect the feasibility and explore the quality of group differences. All calculations on statistical significance presented here are therefore not yet useful for the estimation of the effect sizes, but should weigh the findings of this exploratory trial.

## 3. Results

Twelve boys diagnosed with ADHD and 11 healthy controls, matched for sex, age, height, and weight were included. For results in age, height, and weight see Table 1. Results of the FBB-ADHS questionnaire showed significant group differences for the total score and also for the sub-scores inattention, motor hyperactivity, and impulsivity.

### 3.1. Questionnaire

There was no significant group difference between children with ADHD and the control group, neither for the sum score nor for the subscales Bedtime Resistance, Sleep Anxiety, Night Wakenings, Parasomnias, Sleep-Disordered Breathing, Daytime Sleepiness, Sleep Onset Delay, and Sleep Duration (see Table 2).

### 3.2. Actigraphy

The analysis of the actigraphic data is shown in Table 3 and Table 4 and Figure 2 and Figure 3. The cosinor analysis of activity was performed as groupwise analysis. There was significant circadian rhythmicity detected in both groups (see Table 3), but no significant group difference, neither for average movement (see Figure 2) nor for chronometric parameters (see Table 4). The amount of activity was not significantly different for the groups, neither in the least active 5 h (L5) nor in the most active 10 h period (M10). No other chronometric parameters (inter-daily stability, intra-daily variability, amplitude, acrophase) reached statistical significance (see Table 4). The average means of illumination around 5 p.m. were slightly higher in the ADHD group, but there were no significant differences in the average illumination in the ADHD subjects compared to the controls (see Figure 3).

### 3.3. Hormone Rhythms

The cosinor fits indicated significant 24 h rhythmicity for melatonin and cortisol in both groups (see Table 5).

The percent-rhythms were not significantly different between the groups. Comparing the mesors, acrophases, and amplitudes of the cosinor fits across the groups, we found that the acrophase of the cortisol rhythm was significantly lower for the ADHD group (*p* = 0.002). See Table 6 and Table 7 for further cosinor analyses.

No other cosinor parameter was significantly different between the groups. For the circadian profiles, see Figure 4 and Figure 5. A lower melatonin level during nighttime in the ADHD group (see Figure 4) did not reach a significant level.

### 3.4. Clock Gene Expression Rhythms

On the molecular level, a significant circadian rhythmic expression for both clock genes studied was found in the control group. In contrast, the ADHD group did not display any significantly rhythmic expression for both *BMAL1* and *PER2* (see Table 8).

For the 24 h expression profiles, see Figure 6 and Figure 7. Because no rhythmic expression was to be found in the ADHD group, we did not run through the analysis of the chronometric parameters to compare the groups.

## 4. Discussion

### 4.1. Questionnaire

No significant group differences between the ADHD group and controls were found for the CSHQ sum score and for the subscale scores (see Table 2). This stands in contrast to the results of other studies using the CSHQ in ADHD children. For example, Owens et al. (2000) [10] found significant differences in the sum and nearly all subscale scores in a group of n = 103 (n = 46 control, n = 57 ADHD) children. The authors themselves discussed the comorbidities as confounding factors for their result while we explicitly excluded patients with comorbidities. Palacio-Ortiz et al. (2018) described significantly higher scores (total score, bedtime resistance, sleep onset delay, sleep anxiety, parasomnias, sleep-disordered breathing, daytime sleepiness, night wakings) for a group of n = 228 children with ADHD (n = 142 without medication, n = 86 with medication) and discussed medication as well as comorbidities as important confounding factors [30]. In our study, all subjects were drug naive.

### 4.2. Actigraphy

For motor activity, a significant circadian rhythmicity was found (see Table 3) for both groups. Inter-daily stability (IS) and intra-daily variability (IV) were not significantly different between the groups. This is consistent with the findings of the other study groups [4,20]. Group differences in L5 (the least active 5 h interval), M10 (most active 10 h interval), and other chronometric parameters did not reach the significance levels (see Table 4). This stands in contrast to the results of Boonstra et al. (2007) [4], who found an increased M10 in adults, but in even more contrast with the results of Baird et al. (2012) [20], who found an increased motor activity in the adult ADHD group “across the circadian cycle“ with increased L5, M10, and amplitude in combination with shortened period. In n = 64 children (n = 44 ADHD; n = 22 healthy controls), Dane et al. (2000) did not find significantly different amounts of motor activity between the ADHD and control groups in the morning hours, but rather in the afternoon. In their case-control study of 7 to 12 year old children, no difference was found between the ADHD subtypes [41]. Imeraj et al. (2011) did not find an increased nocturnal motor activity in their study on n = 60 children (n = 30 ADHD; n = 30 healthy controls) [42]. All of these studies were conducted using wrist worn devices the same as the one we used. Faedda et al. (2016) used a belt-worn actigraph instead, and found significant differences in L5 and RA between the non-medicated ADHD and healthy controls in their case control study on children and adolescents (5 to 18 years) [43]. Ironside et al. (2010) described a significant phase delay and a reduced RA in ADHD children when treated with methylphenidate [44]. In their meta-analysis including 393 patients, Crescenzo et al. (2014) described actigraphy as a useful and effective tool to measure the effects of medication in ADHD children [9]. In our cohort, all participants were drug naive. Regarding the light exposure, Baird et al. (2012) described an increased average illumination during night time until the morning hours in the adult ADHD group [20]. We found a slight difference in the evening hours around 5 p.m. with more light in the ADHD group (see Figure 3), but group differences did not reach a significant level in our children population. Again, a possible influence in the small sample size in our cohort must be taken into account. Another influence factor might be that our population was drug naive and without any comorbidity. Beyond that, there could be a difference between the children and adults. Most importantly, actigraphy does not delineate between endogenous circadian processes and environmental factors. In our cohort, actigraphic parameters were not capable of distinguishing ADHD patients from the healthy control. In their meta-analysis about sleep and ADHD in childhood, Díaz-Román et al. (2016) included eight studies using objective tools (ADHD: n = 133 with PSG and n = 58 with actigraphy; controls: n = 169 PSG and n = 92 actigraphy). The authors discussed several confounding factors as reasons for the inconsistence in study results and provided recommendations for the study design and sample characteristics [7].

### 4.3. Melatonin and Cortisol

In our dataset, a robust rhythmicity in the melatonin and cortisol profiles for both groups was shown (Table 5). This stands in contrast to prior findings in ADHD adults, where a loss of rhythmicity for melatonin was described [20]. The peak of melatonin concentration in our study was found at the same time in both groups. Similar to the findings of Baird et al. (2012), the melatonin profile in our cohort showed a damped amplitude in the ADHD group (see Figure 4). However, in our dataset, the amplitude difference did not reach a significant level (see Table 7). Chronometric parameters were not significantly different between both groups (see Table 6 and Table 7). These results are consistent with those found by Baird et al. (2012) [20]. Molina-Carballo et al. (2013) stated that children with ADHD showed higher morning melatonin levels, which disappeared after the administration of methylphenidate [45]. As noted earlier, all children in our study were drug naive. Cubero-Millán et al. (2014) described different melatonin levels in ADHD subtypes with higher levels in the hyperactive-impulsive subgroup and lower levels in the inattentive subgroup. Furthermore, they reported that medication with methylphenidate seems to lower melatonin levels in their prospective open quasi-experimental study of children aged 5–14 years by measuring the blood melatonin and urinary 6-sulfatoxymelatonin [46]. Nováková et al. (2011) reported no difference in the nocturnal melatonin levels between the ADHD and control groups, but described an altered melatonin rhythm profile in older ADHD children (n = 77, aged 6 to 12 years) [19]. Paclt et al. (2011) described alterations in the melatonin rhythm profiles in ADHD children with a comorbid conduct disorder in their study in n = 88 children (n = 34 ADHD, n = 43 controls, n = 11 anxiety) aged from 6 to 12 years [47].

For cortisol, we found significant cosinor fits in the ADHD and control groups (see Table 5), similar to Baird et al. (2012) [20]. In our study, the acrophase of the cortisol profile was significantly lower in ADHD, which indicates a phase shift of the peak concentration 4 h earlier than in the healthy control group (see Table 6 and Figure 5). Interestingly, Baird et al. (2012) also found a phase delay for the cortisol peak, but in their study, the acrophase of the cortisol rhythm was significantly earlier for the ADHD group [20]. For children, Isakson et al. demonstrated that ADHD showed lower awakening responses in their case control study in n = 81 ADHD children and n = 88 healthy controls [18]. Imeraj et al. found no differences in the morning cortisol levels, but steeper declines during day in n = 66 children (33 ADHD, 33 controls), aged 6–11 years [17]. Hirvikoski et al. (2009) did not find any differences in diurnal rhythm or level of salivary cortisol in their adult cohort (n = 28 ADHD, n = 28 healthy controls) [48]. In sum, the results remain inconsistent. Probands age, comorbidity, and medication seem to have an impact on the melatonin secretion profile. The same confounding factors were discussed by Baird et al. (2012) [20]. However, in our cohort, the rhythmic profiles of both melatonin and cortisol were not sufficient as markers for ADHD.

### 4.4. Clock Gene Expression Rhythms

There have been several studies on clock genes that have focused on structural changes in the DNA such as single-nucleotide polymorphisms (SNIPs). One SNIP of the circadian locomotor output cycles protein kaput (*CLOCK*) gene rs1801260 was investigated in different cohorts with controversial results. Kissling et al. (2008) assessed 143 adult subjects in a German population and found a significant relation between this SNIP and ADHD [23], as did Xu et al. (2010) for a Taiwanese (n = 212), but not for a British (n = 180) population of ADHD children aged from 5 to 15 years [24]. In 2020, Wang et al. performed a pathway-based analysis in n = 168 ADHD children and n = 233 healthy controls in a Chinese population and showed that the combination of different SNIPs within the clock gene system was associated with a significantly higher risk for ADHD in children [25]. Baird et al. (2012) were the first to examine the regulation of clock gene expression in ADHD in 2012. The minimal invasive method used to obtain the material was well-tolerated by all subjects and enabled clock gene expression investigations outside laboratory conditions by the families themselves. It therefore allows for a naturalistic setting. The Baird study group found a rhythmic expression of *BMAL1* and *PER2* in the control, but not in the ADHD group [20]. This was similar to our results (see Table 8). Because there was no rhythmic expression to be found for both genes in the ADHD group, we did not further analyze the chronometric parameters.

As noted by Baird et al. (2012), the SCN is the cerebral structure integrated in the circadian regulation of physiology and behavior [20]. We measured the oral mucosa instead of the SCN material, and there was no information on how oscillations of clock gene expression in SCN and in the peripheral cells would correlate. This is why we decided not to calculate the possible correlations with the clinical parameters of ADHD. However, it is noteworthy that the alteration of this rhythmic expression of clock genes in peripheral cells is to be found in children and adults with ADHD. Thus, DNA structural analyses are inconsistent for clock genes in ADHD, but analyses of the circadian rhythmicity of clock gene expression appear to be specific and sensitive (with only a small cohort like ours) and independent of age.

## 5. Limitations and Conclusions

In summary, we did not find significant groupwise differences in the sleep questionnaires’ total score and its subscores. Actigraphic parameters did not show significant differences between the groups. There were slight differences in the secretion profiles of melatonin and cortisol. The clearest difference between both groups according to the circadian parameters were found in the alterations of the expression profiles of two clock genes, *BMAL1* and *PER2*. These alterations seem to be robust markers for ADHD in both children and adults. In relation to the recommendations for the study design and sample characteristics of Dáz-Román et al. (2016) [7], our setting combined subjective and objective tools to measure sleep and circadian rhythmicity. Both groups were controlled for medical and psychological problems. Through our extensive inclusion procedure, we only included properly diagnosed ADHD patients and can therefore guarantee the absence of false positives. As mentioned earlier, the study presented here is a pilot study. General conclusions about the population are not possible, or only to a very limited extent, because of the small sample size in our study. Subgroup analyses with a focus on hyper motor activity/impulsiveness, and especially aggressiveness/social behavior disorder, would be interesting [49,50], but are statistically not advised in our small cohort.

In our study, we regarded the clock gene expression profiles to be sensitive markers to distinguish ADHD patients from the healthy controls. This was similar to the results of Baird et al. (2012) in adults [20]. Are clock gene profiles therefore a sufficient marker for ADHD for children and adults? There are study groups currently developing a procedure to measure the circadian profile of clock gene expression via one blood sample [51]. Can clock gene profiles become future markers in ADHD diagnostics using one blood draw? Publications from animal models to humans illustrate the impact of non-stimulant and stimulant medication on circadian rhythmicity [52,53,54,55]. One promising way of future research would therefore be to examine the impact of medication on the connection of genetic and clinical expression.

## Figures and Tables

**Figure 1 brainsci-12-01198-f001:**
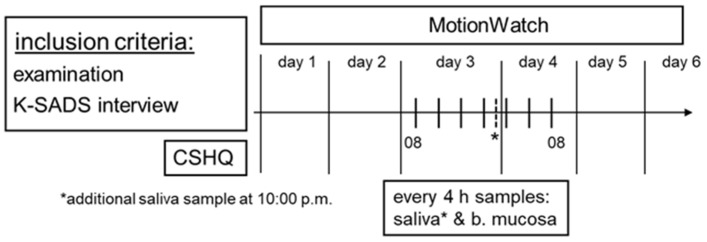
The study protocol, h = hours, K-SADS = Schedule for Affective Disorders and Schizophrenia for School-Age Children-Present and Lifetime Version, CSHQ = Children Sleep Habits Questionnaire.

**Figure 2 brainsci-12-01198-f002:**
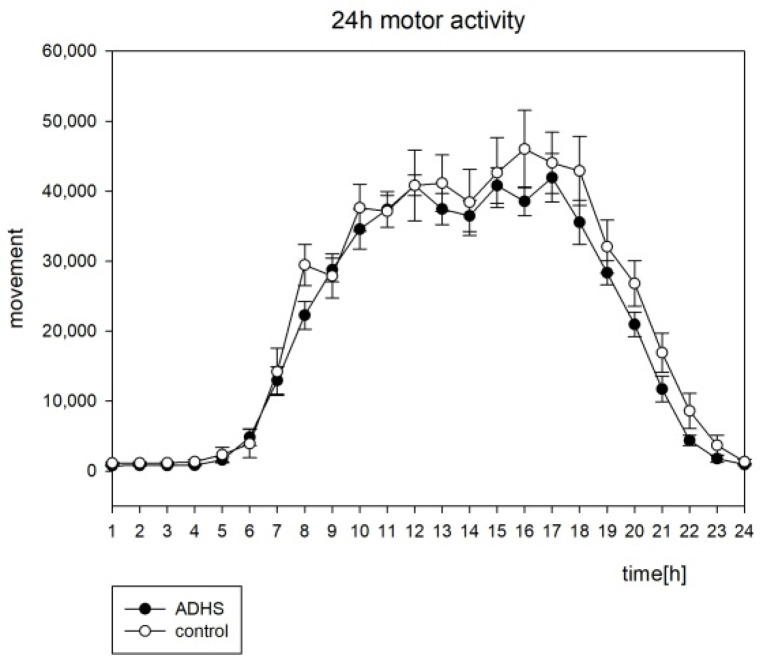
The circadian profile of motor activity with standard error, h = hours.

**Figure 3 brainsci-12-01198-f003:**
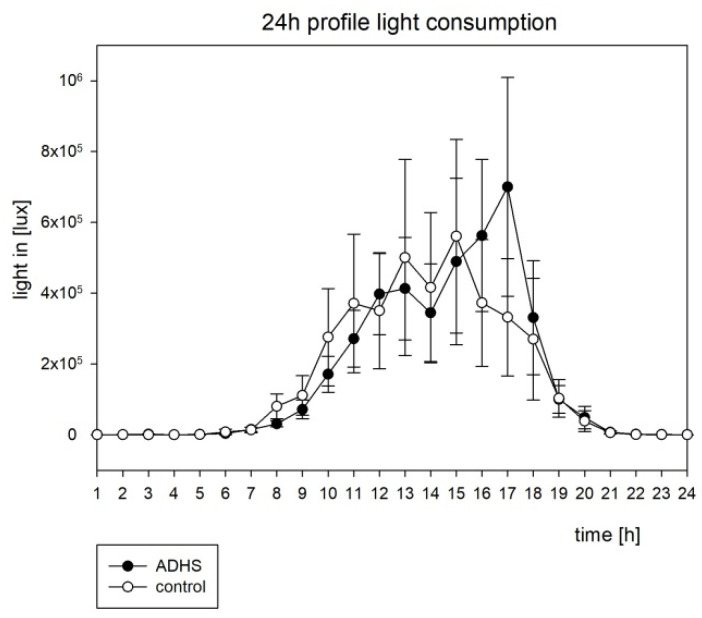
The circadian profile of illumination with the standard error, h = hour.

**Figure 4 brainsci-12-01198-f004:**
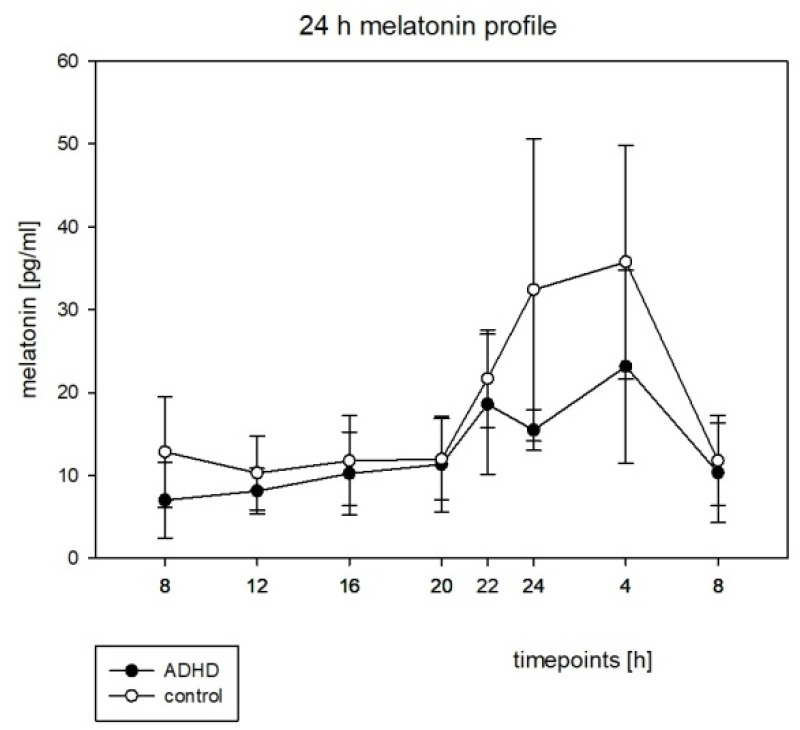
The circadian profile of saliva melatonin with the standard error, h = hours, pg = picogram, mL = milliliters.

**Figure 5 brainsci-12-01198-f005:**
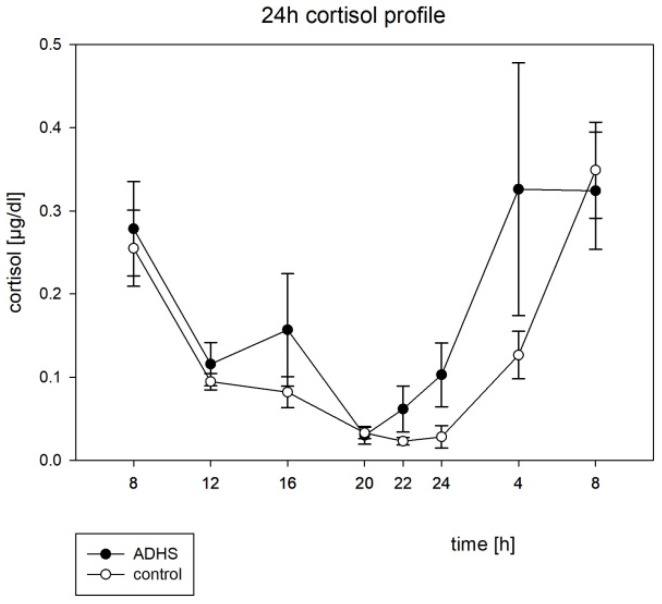
The circadian profile of the saliva cortisol with the standard error, h = hours, µg = microgram, dL = deciliter.

**Figure 6 brainsci-12-01198-f006:**
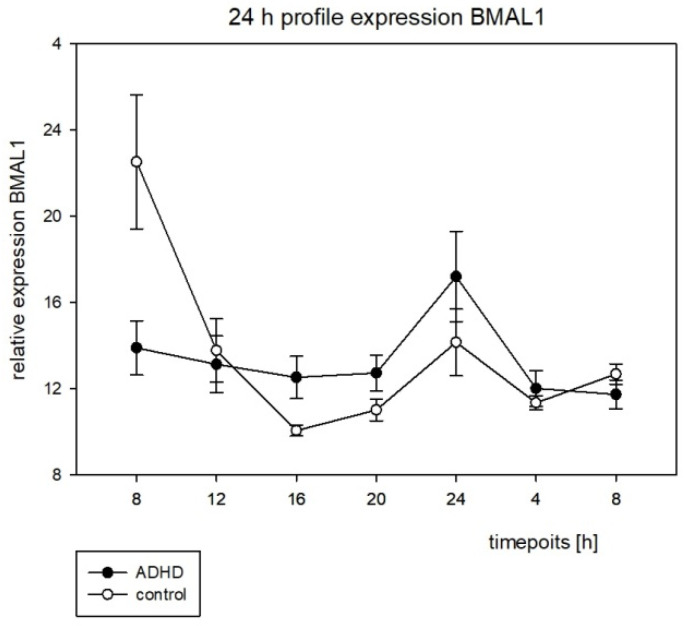
The circadian profile of the BMAL1 expression with the standard error, h = hours.

**Figure 7 brainsci-12-01198-f007:**
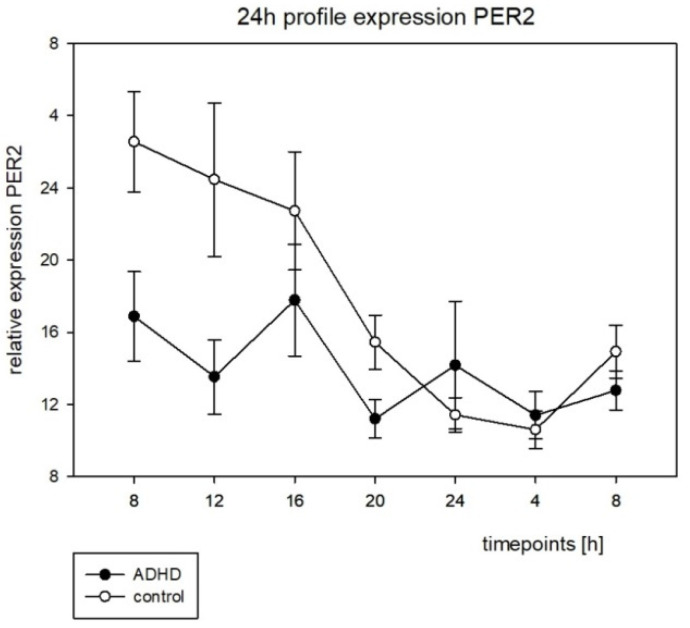
The circadian profile of the PER2 expression with the standard error, h = hours.

**Table 1 brainsci-12-01198-t001:** The descriptive statistics. IQ = intelligence quotient, sd = standard deviation, cm = centimeter, kg = kilogram, n = number.

Group	N	IQ (Mean ± SD)	Age (Mean ± SD) [Months]	Height (Mean ± SD) [cm]	Weight (Mean ± SD) [kg]
ADHD	12	95.42 ± 7.48	117 ± 14.76	144.13 ± 9.86	40.48 ± 13.25
control	11	103.73 ± 8.36	115.36 ± 15.01	145.00 ± 11.00	36.15 ± 15.01

**Table 2 brainsci-12-01198-t002:** The groupwise comparison of CSHQ scores. sd = standard deviation. The Mann–Whitney U-test was used for testing the group effects.

	Control (Mean ± SD)	ADHD (Mean ± SD)	Z; *p*
Sum score	43.0 ± 3.57	40.18 ± 5.98	1.55; 0.12
Subscores			
Bedtime resistance	7.09 ± 2.17	6.75 ± 1.06	0.14; 0.89
Sleep onset delay	1.18 ± 0.41	1.58 ± 0.79	1.33; 0.18
Sleep duration	3.36 ± 0.67	4.50 ± 2.07	1.16; 0.25
Sleep anxiety	4.64 ± 1.80	4.58 ± 1.00	0.72; 0.47
Parasomias	8.36 ± 1.29	8.17 ± 1.992	0.25; 0.80
Sleep-disordered breathing	3.18 ± 0.60	3.17 ± 1.19	0.76; 0.45
Daytime sleepiness	11.18 ± 2.86	12.33 ± 2.87	0.93; 0.35
Night wakings	3.45 ± 0.69	4.00 ± 1.50	0.86; 0.39

**Table 3 brainsci-12-01198-t003:** The cosinor analysis movement, * = significant (*p* ≤ 0.05).

	% Rhythm	*p*-Value
ADHD	90.17	<0.0001 *
control	86.64	<0.0001 *

**Table 4 brainsci-12-01198-t004:** The chronometric parameters movement (actigraphy). The Mann–Whitney U-test was used for testing the group effects, sd = standard deviation.

	ADHD (Mean ± SD)	Control (Mean ± SD)	Z; *p*
Inter-daily stability (IS)	0.617 ± 0.04	0.601 ± 0.11	0; 1
Intra-daily variability (IV)	0.571 ± 0.09	0.623 ± 0.10	1.2; 0.23
M10	37,608 ± 5182	40,812 ± 11,655	1.42; 0.16
L5	794 ± 220	1074 ± 445	1.42; 0.16
Amplitude	22,758 ± 3294	24,239 ± 7373	0. 74; 0.46
Acrophase	−210.33 ± 10.18	−212.73 ± 13.81	1.39; 0.165

**Table 5 brainsci-12-01198-t005:** The cosinor analysis for hormones (cortisol + melatonin), * = significant (*p* ≤ 0.05).

	% Rhythm	*p*-Value
Melatonin		
ADHD	36.22	0.0007 *
Control	53.50	0.0302 *
Cortisol		
ADHD	49.63	0.0046 *
Control	66.31	0.0003 *

**Table 6 brainsci-12-01198-t006:** The chronometric parameters of the cortisol profile; * = significant (*p* ≤ 0.05).

	ADHD	Control	*p*-Value
Mesor	0.121730139	0.16991	0.272
Amplitude	0.123209832	0.134743	0.918
Acrophase	−2.136710329	−1.77627	0.002 *

**Table 7 brainsci-12-01198-t007:** The chronometric parameters of the melatonin profile.

	ADHD	Control	*p*-Value
Mesor	17.75686207	12.66987	0.544
Amplitude	12.831147	6.280137	0.696
Acrophase	−0.490405853	−0.30878	0.196

**Table 8 brainsci-12-01198-t008:** The cosinor analysis clock genes (BMAL1, PER2), * = significant (*p* ≤ 0.05).

	% Rhythm	*p*-Value
BMAL1		
ADHD	37.15	0.277
Control	38.85	0.004 *
PER2		
ADHD	38.48	0.580
Control	48.22	0.007 *

## Data Availability

The data presented in this study are available on request from the corresponding author. The data are not publicly available due to privacy and ethical restrictions.

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
