# Peer review of "Clock Genes Profiles as Diagnostic Tool in (Childhood) ADHD—A Pilot Study"

_brainsci, 2022, doi:10.3390/brainsci12091198_

Round 1

Reviewer 1 Report

ID: brainsci-1892695

Title: Clock Genes profiles as diagnostic tool in (childhood) ADHD - A pilot study.

Thank you for providing a chance to review this manuscript.

Comment: minor revision.

Detailed information:

Title

Line 2-3, page 1: Why do you use "(childhood)"?

1. Introduction

Line 58-61, page 2: What is your study hypothesis? What are the clinical implications of conducting this study? Why is the format of this paragraph different from the previous one?

2. Materials and Methods

Line 79-86, page 2: Is this questionnaire the original version in this study? What is its reliability and validity? If not, has the version you used been adequately validated?

3. Results

Line 172, page 4: Is this a sufficient sample size? Any reference?

Line 178, page 5: Please report the reliability of this questionnaire.

Tables: Three-line tables are needed. Don't mix up the table name and note, please modify them. And standardize the number of decimal places in the tables.

4. Discussion

Line 257-258, page 10: I suggest that this phrase could be placed in "limitations".

5. Conclusions and Limitations

Given your content, I suggest changing the subheading to "Limitations and Conclusions”. You have done massive work, showing the strengths to the readers would never harm.

It can be seen that a lot of work was carried out in this study, but some issues I mentioned may need to be answered and addressed.

Thank you and my best,

Your reviewer

Author Response

Dear Reviewer I, please see the attachment. 

Reviewer 2 Report

dear authors

thank you for the paper

minor points: 

1) line 31-32 provide 95%ci and pay attention to . or , all tables used , pls correct to .

2)line 33 provide summary using meta-analyses some very important literature missed e.g. crescenzo and diaz Roman

pls provide better synopsis

3)line 61 pls provide explicit objective

4)lines 69-71 explain more conners used? type of adhd

major concerns:

1)methods missing sample, sample size calc, inclu and exlu criteria and source of recruitment for cases and controls. it's very vague to say other comorbidities as exclusion criteria

2)actigraphy setting/calibration for peadiatric need to be mentioned by whom 

3) statistical analysis inappropriate to use parametric tests with such small sample size according to table 1 and 2 data clearly not normally distribution

4)results difficult to explain as it mainly against what we already know and I can only think it's that psychostimulatns masking the results confounded - see my comment above #1 we need better reading of methods to judge this study further

Author Response

Dear reviewer 2, please see the attachment.

Round 2

Reviewer 2 Report

thank you for addressing my concerns.

kindly line 32 delete the (no CI provided).